# Ultralow-voltage operation of light-emitting diodes

Yaxiao Lian[1,5], Dongchen Lan [2,3,5], Shiyu Xing[1,5], Bingbing Guo[1], Zhixiang Ren[1], Runchen Lai [1], Chen Zou[1], Baodan Zhao [1,4], Richard H. Friend [4] & Dawei Di [1,4✉]

For a light-emitting diode (LED) to generate light, the minimum voltage required is widely considered to be the emitter's bandgap divided by the elementary charge. Here we show for many classes of LEDs, including those based on perovskite, organic, quantum-dot and III–V semiconductors, light emission can be observed at record-low voltages of 36–60% of their bandgaps, exhibiting a large apparent energy gain of 0.6–1.4 eV per photon. For 17 types of LEDs with different modes of charge injection and recombination (dark saturation currents of ~$10^{-39}$–$10^{-15}$ mA cm$^{-2}$), their emission intensity-voltage curves under low voltages show similar behaviours. These observations and their consistency with the diode simulations suggest the ultralow-voltage electroluminescence arises from a universal origin—the radiative recombination of non-thermal-equilibrium band-edge carriers whose populations are determined by the Fermi-Dirac function perturbed by a small external bias. These results indicate the potential of low-voltage LEDs for communications, computational and energy applications.

[1] State Key Laboratory of Modern Optical Instrumentation, College of Optical Science and Engineering; International Research Center for Advanced Photonics, Zhejiang University, Hangzhou 310027, China. [2] College of Electrical Engineering, Zhejiang University, Hangzhou 310027, China. [3] Australian Centre for Advanced Photovoltaics, University of New South Wales, Sydney 2052, Australia. [4] Cavendish Laboratory, University of Cambridge, JJ Thomson Avenue, Cambridge CB3 0HE, United Kingdom. [5] These authors contributed equally: Yaxiao Lian, Dongchen Lan, Shiyu Xing. ✉email: daweidi@zju.edu.cn

The development of LEDs[1–6] has created far-reaching impacts on lighting, display and information industries. Emerging LED technologies, including organic LEDs (OLEDs)[4–8], quantum-dot LEDs (QLEDs)[9–11], and perovskite LEDs (PeLEDs)[12–18], are gaining significant attention due to their promise as next-generation light sources. The key mechanism responsible for the light emission from LEDs is electroluminescence (EL), the radiative recombination of injected electrons and holes under an external voltage. It has been suggested that the minimum (threshold) driving voltage required to create photons from the EL process equals to the bandgap ($E_g$) of the emissive material divided by the elementary charge ($q$), in consideration of the energy conservation principle[19,20], while free energy considerations allow a marginal energy gain of a few $kT$ (where $k$ is the Boltzmann constant and $T$ is the temperature)[21]. Studies suggested that the minimum voltage may be reduced through various mechanisms, including thermally assisted upconversion[22–26], sequential charge injection[27], interfacial dipoles[28], triplet-triplet annihilation upconversion (TTA-UC)[29,30], and Auger processes[10,31–34]. Recently, an operating voltage of as low as 77% of the nominal bandgaps was observed for LEDs based on III–V semiconductors, and this was attributed to enhanced radiative recombination enabled by a novel quantum well design[35]. For OLEDs, a minimum operating voltage of $\sim 0.5E_g/q$ was reported[30,31,33], though whether a TTA process could be used to explain the origin of this low operating voltage is still a subject of debate[30,33,36]. Sub-bandgap operating voltages were also observed for perovskite[13], and quantum-dot LEDs[10,11,26,27,32,37,38] (Supplementary Table 1). These observations lead to an open question of what the lowest possible driving voltages really are for electroluminescence, and whether they stem from the same origin. Ultralow-voltage operation of LEDs may create new opportunities for next-generation energy-efficient optoelectronics.

In this work, using high-sensitivity photon counting measurements on 17 types of LEDs, we show that EL at voltages significantly below the emitter's bandgap is possible for many classes of LEDs, and is not exclusive to a few novel material systems. The similar shapes of the EL-voltage curves reveal a universal origin of ultralow-voltage operation, in spite of the very different modes of charge injection and recombination in these devices.

## Results

We began our investigation by measuring the minimum operating voltages of LEDs based on emerging material systems. Perovskite LEDs were our first experimental subjects. We fabricated iodine-based near-infrared "FPI"[14,39], "NFPI"[40] and bromide-based green-emitting "PCPB"[41] perovskite LEDs with peak EQEs of ~10% (Fig. 1a–c and Supplementary Fig. 1a–c. See Methods for fabrication details). We observed that, for these perovskite LEDs, the minimum voltages for EL were 1.3 V, 1.3 V and 1.9 V (Fig. 1a to c), while the EL peak photon energies were 1.55 eV, 1.56 eV and 2.4 eV (Supplementary Fig. 2a, b and Supplementary Fig. 3b), respectively (the minimum detectable photon flux is $\sim 10^{16}\,s^{-1}\,m^{-2}$ for our standard measurement setup, see Methods for details). Here, EL peak photon energies are used to provide conservative estimates of the bandgaps. The minimum operating voltages observed were 83%, 83% and 79% of the bandgaps for FPI, NFPI and PCPB PeLEDs, respectively. The observation of near- or sub-bandgap operating voltages for these LEDs is consistent with recent findings for efficient PeLEDs[12,13,39,42,43]. We performed similar experiments for phosphorescent OLEDs based on tris(2-phenylpyridine)iridium(III) (Ir(ppy)$_3$) and bis[2-(4,6-difluorophenyl)pyridinato-C2,N](picolinato)iridium (FIrpic)[44], thermally activated delayed fluorescence (TADF) OLEDs based on 1,3,5-tetrakis(carbazol-9-yl)-4,6-dicyanobenzene (4CzIPN)[7], polymer OLEDs based on poly(9,9-dioctylfluorene-alt-benzothiadiazole) (F8BT)[45], fluorescent small-molecule OLEDs based on rubrene[31], and II–VI chalcogenide QLEDs based on CdSe/ZnS quantum dots[46]. Sub-bandgap voltage EL was similarly observed (Fig. 1d to h).

Further, we measured commercial III–V LEDs based on GaN, AlGaP, GaP, GaAsP, InAlGaP, AlGaAs, GaAs, and InGaAsP. Sub-bandgap voltage operation was similarly observed. For GaAsP-based LEDs with an $E_g$ of 1.95 eV (Fig. 1i), EL could be clearly observed under an applied voltage of 1.45 V ($0.74E_g$) using the same measurement setup (See Methods for details). Importantly, the EL spectra remained unshifted as the driving voltages varied from above to clearly below the bandgaps (Fig. 2 and Supplementary Figs. 2, 3), while sub-bandgap voltage EL is shown to be a general phenomenon.

A key scientific question is what the minimum voltages really are for the operation of LEDs. To find an answer to this problem, we employed a highly sensitive photon detection system (Supplementary Fig. 4) for the determination of the onset of EL, greatly enhancing the measurement sensitivity for weak photon emission (minimum detectable photon flux is $\sim 10^9\,s^{-1}\,m^{-2}$, which is 7 orders of magnitude more sensitive than the standard measurement setup; see Materials and Methods for details). EL was detected from our perovskite LEDs at voltages equivalent to 0.4–0.6 $E_g$ (Fig. 3a), representing the lowest driving voltages reported for perovskite LEDs to date. For FPI, NFPI and PCPB PeLEDs emitting at ~800 nm, ~790 nm and ~515 nm, the minimum voltages for observing EL were ~0.86 V, ~0.72 V and ~1.52 V, corresponding to $qV_m/E_g$ of ~55%, ~46% and ~63% respectively (Fig. 3a). Here, $V_m$ denotes the measured minimum voltage required for generating detectable EL. The apparent energy gap $\Delta E = E_g - qV_m$, was as large as 0.9 eV. This is more than an order of magnitude greater than the room-temperature thermal energy ($kT = 26$ meV at 300 K). Using bandpass filters with cut-off wavelengths close to the materials' bandgaps (Supplementary Fig. 5), we confirmed that these photons did not arise from the recombination of sub-bandgap states.

We similarly observed minimum operating voltages of 1.75 V ($0.73E_g/q$) and 1.9 V ($0.72E_g/q$) for phosphorescent OLEDs based on Ir(ppy)$_3$ and FIrpic, 1.8 V ($0.77E_g/q$) for TADF OLEDs based on 4CzIPN, 0.8 V ($0.36E_g/q$) for fluorescent small-molecule OLEDs based on rubrene, 1.6 V ($0.74E_g/q$) for polymer OLEDs based on F8BT, 1.1 V ($0.56E_g/q$) for II–VI QLEDs based on CdSe/ZnS QDs, and 1.7 V ($0.63\ E_g/q$), 1.4 V ($0.65E_g/q$), 1.3 V ($0.62E_g/q$), 1.25 V ($0.6E_g/q$), 0.78 V ($0.5E_g/q$), 0.72 V ($0.48E_g/q$), 0.60 V ($0.43E_g/q$) and 0.54 V ($0.42E_g/q$) for inorganic LEDs based on GaN, AlGaP, GaP, GaAsP, InAlGaP, AlGaAs, GaAs and InGaAsP, respectively (Fig. 3b to d). Record-low operating voltages were found for each class of LEDs (Fig. 3e). We noted that the apparent energy gaps ($\Delta E$) were on the order of ~1 eV. A summary of measured minimum voltages and $\Delta E$ values we observed is provided in Table 1, and the measurements were reproducible across a number of devices (Supplementary Fig. 6). Our experiments collectively demonstrate that the EL operation at sub-bandgap voltages is a universal phenomenon across different classes of LEDs, and the operating voltages can reach values of $\sim 0.5E_g/q$ or below. For NFPI perovskite devices, rubrene devices and commercial inorganic NIR devices, the measured $V_m$ values were even below the threshold voltage limits set by the TTA/Auger processes (Fig. 3f). These cannot be explained by previously published mechanisms[23–25,27–29,31–34]. The lower EQE values at low voltages (Supplementary Fig. 7) arise from the larger fraction of non-radiative recombination losses typically observed in semiconductors under low injection conditions[35]. Our measurements in Fig. 3e show that there is still a large number of photons being emitted under ultralow voltages, suggesting that

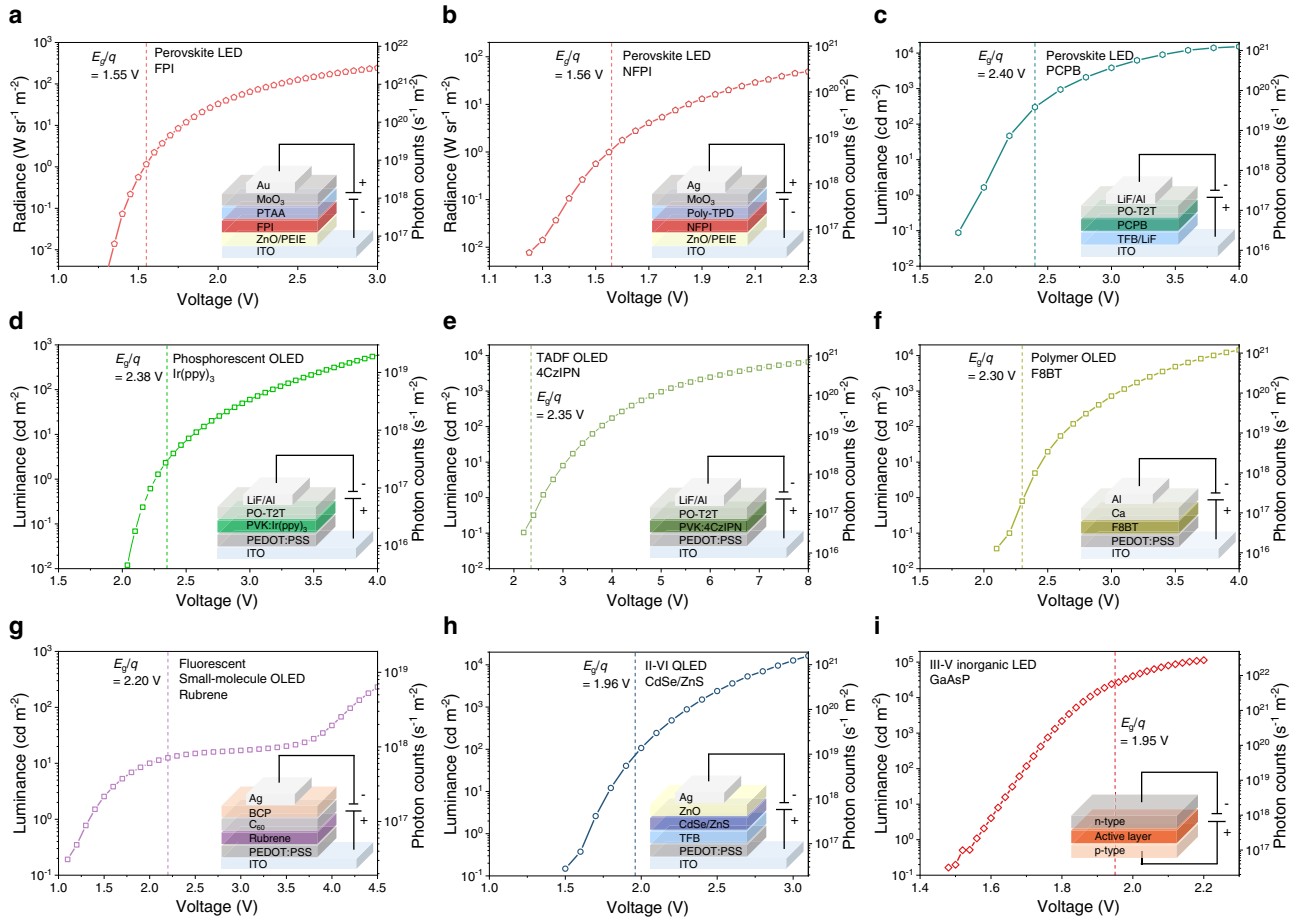

**Fig. 1 EL intensity-voltage characteristics of different classes of LEDs. a** Near-IR-emitting FAPbI₃ (FPI) perovskite LED. **b** Near-IR-emitting NFPI perovskite LED. **c** Green PCPB perovskite LED. **d** Phosphorescent OLED based on Ir(ppy)₃. **e** TADF OLED based on 4CzIPN. **f** Polymer OLED based on F8BT. **g** Fluorescent small-molecule OLED based on rubrene. **h** II–VI QLED based on CdSe/ZnS QDs. **i** Commercial III–V inorganic LED based on GaAsP. The bandgaps for each emissive material are marked by dashed lines. Insets are schematics of the respective LED device structures.

the $V_m$ values may be reduced further by improving the instrumental sensitivity.

The current–voltage curves of various classes of LEDs exhibit very different characteristics (Supplementary Fig. 8a). However, all LEDs show remarkably similar EL intensity–voltage behaviours under low operating voltages (Fig. 3e), and follow the conventional diode equation described below as Eq. (1), where the light emission tracks the current density, $j$.

$$j = j_0 \left[ e^{\frac{q(V - jR_s)}{nkT}} - 1 \right] \qquad (1)$$

where $j_0$ is the diode dark saturation current density ($j_0$ is negatively correlated with $E_g$ in a general form of $j_0 = \Lambda e^{-\frac{E_g}{kT}}$, where $\Lambda$ is related to material properties. See Supplementary Note 1 for details), $n$ is the ideality factor, $k$ is the Boltzmann constant, $T$ is the temperature, and $V$ is the external voltage applied across the diode with mimimum influence from series resistance ($R_s$) at low voltages. While Eq. (1) is generally derived for unipolar p–n junction diodes, we see here that it clearly models the electron-hole recombination current for these diodes irrespective of the choice of semiconductor materials and charge-injection electrodes. In essence, the EL intensity ($I_{EL}$) is linked to the current density via the external quantum efficiency (EQE) of the LED (Eq. (2)).

$$I_{EL} = EQE \frac{j}{q} \qquad (2)$$

The relation between the EL intensity and the applied voltage can be described by Eq. (3) (See Supplementary Note 2 for details).

$$\log(I_{EL}) = \frac{\log(e)q}{nkT}V + \log(EQE) - \log(e)W\left(\frac{qR_s j_0}{nkT}e^{\frac{qV}{nkT}}\right) + \log\left(\frac{j_0}{q}\right) \qquad (3)$$

in which W is the Lambert W-function[47]. EQE can be determined experimentally from the current–voltage and EL intensity-voltage data. The EL intensity-voltage characteristics of our LEDs could be nicely fitted by this preliminary model (Supplementary Fig. 8).

We note that the dark saturation current density ($j_0$) varies greatly across different classes of LEDs, from ~$10^{-39}$ mA cm⁻² for FIrpic, ~$10^{-35}$ mA cm⁻² for PCPB perovskite, ~$10^{-27}$ mA cm⁻² for F8BT, ~$10^{-25}$ mA cm⁻² for CdSe/ZnS to ~$10^{-15}$ mA cm⁻² for InGaAsP devices (Fig. 3g and Supplementary Table 2). $j_0$ contains important information on the physics of charge recombination in LEDs, and it depends on $E_g$ and $\Lambda$ (Supplementary Note 2). The measured $j_0$ exhibit a negative correlation with $E_g$. To allow a clearer comparison across different material systems, we also calculated the bandgap-weighted dark saturation current densities ($j_0 e^{\frac{E_g}{kT}}$) for the LEDs we measured (Fig. 3h). Interestingly, the 'weighted $j_0$' values of the perovskite LEDs based on FPI, NFPI and PCPB now approaches the regime for III–V semiconductor LEDs

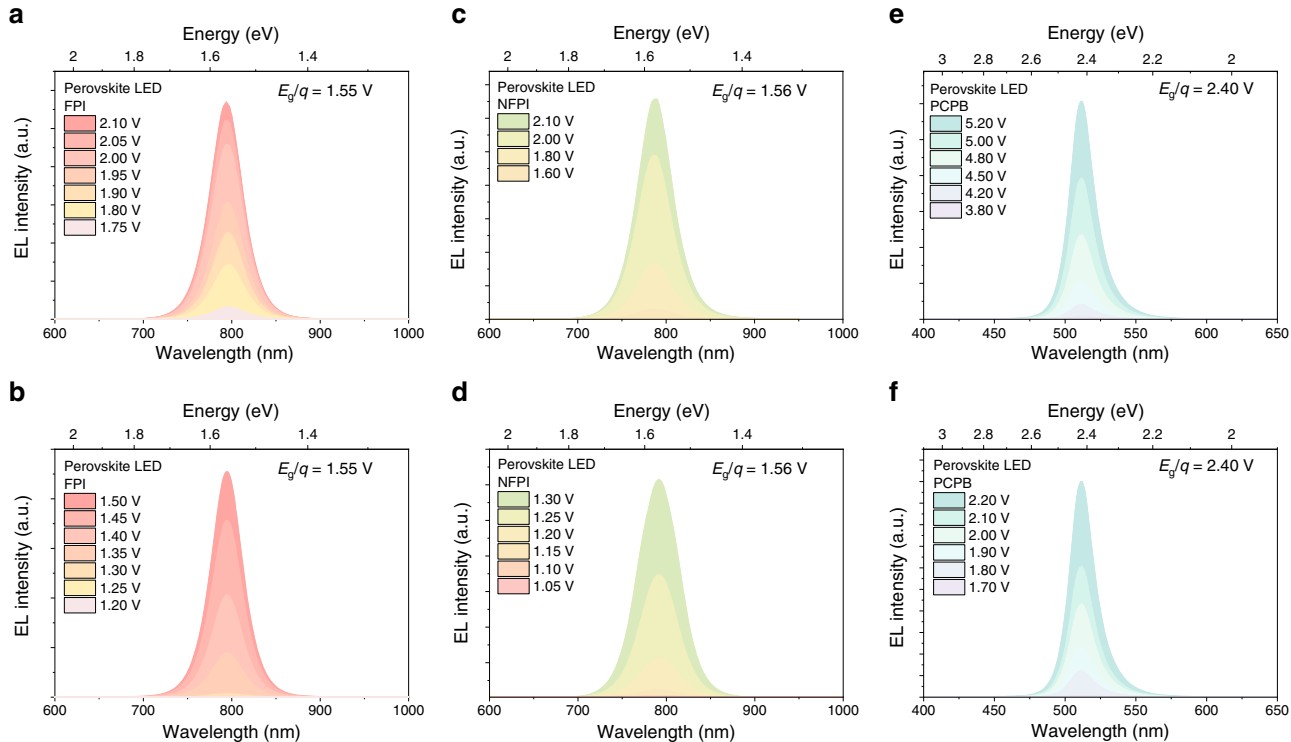

**Fig. 2 EL spectra of perovskite LEDs under above- and sub-bandgap voltages. a, b** EL spectra of a FPI perovskite LED driven at different bias above and below the bandgap voltage. **c, d** EL spectra of a NFPI perovskite LED driven at different bias above and below the bandgap voltage. **e, f** EL spectra of a PCPB perovskite LED driven at different bias above and below the bandgap voltage.

(Fig. 3h and Supplementary Table 2). The ideality factors presented in Supplementary Table 2 fall within the range between 1 and 2. Smaller $n$ (closer to 1) suggests a reduced fraction of defect-assisted recombination, generally corresponding to higher-quality diodes with efficient charge transport and radiative recombination. This is the case for LEDs based on III–V semiconductors and PeLEDs based on FPI. The ideality factors for most of the PeLEDs, QLEDs and OLEDs tested are slightly higher, likely due to the reason that the charge transport and recombination processes are influenced by traps[48,49].

We plotted $V_m$ versus $j_0$ and they appear to be negatively correlated across many classes of LEDs (Fig. 3i). Such correlation could be understood using Eqs. (1) and (2). The EL intensity ($I_{EL}$) is directly related to the current density ($j$) through EQE, and $j$ is determined by $j_0$. Increasing $j_0$ may raise EL intensity at the same applied voltage, reducing the apparent threshold voltage for the same photon flux. Among the many factors affecting $j_0$ (Supplementary Note 1), the emissive material's bandgap $E_g$, on which $j_0$ is exponentially dependent, plays a significant role (as evidenced in Fig. 3g). As such, LEDs based on materials with smaller $E_g$ normally have larger $j_0$ (presuming other factors such as the trap states have smaller influence), leading to the normally negative correlation between $j_0$ and $V_m$. However, larger $j_0$ may also arise from a higher density of defect states, particularly for LEDs based on the same or similar emissive materials, in which case the same $V_m$-$j_0$ correlation may no longer hold, as $V_m$ is negatively correlated with the emission efficiency (see e.g. Supplementary Fig. 9). In a related case of imbalanced charge injection where one type of charge carriers (either electrons or holes) dominates over the other, $V_m$ may not show a negative correlation with $j_0$ (see e.g. Supplementary Fig. 10). This is because $j_0$ may partly originate from trap-assisted non-radiative recombination (in the bulk or at the charge-transport interfaces), which cannot contribute to the EL. It is worth noting that the

models originally developed for conventional inorganic semiconductors can also be used to describe the general behaviour of emerging classes of LEDs with vastly different material properties, pointing toward a universality in the operating principles of different LEDs.

To gain further insight, we employ a widely used device simulation software "Setfos"[50,51] to model the emission behaviour of LEDs. We constructed model devices including a lead-iodide perovskite LED, and a standard phosphorescent OLED based on Ir(ppy)$_3$ (Fig. 4) (See Methods and Supplementary Tables 3, 4 for details). The simulation results (Fig. 4c, d) show that both types of LEDs are capable of generating EL at voltages well below the bandgaps in a fashion similar to what we have observed with our experimental setup (Fig. 3), consistent with the LED model we described. From the simulations, it can be seen that the intensities of the output photon fluxes correlate strongly with the densities of injected charges. At operating voltages significantly below the bandgap, there are appreciable levels of electron and hole populations contributing to the radiative recombination (Fig. 4e, f). These results are consistent with the conventional diode law and with our proposed mechanism for sub-bandgap EL. At similar photon fluxes, the modelled Ir(ppy)$_3$ OLED operates at higher voltages in comparison to that of the perovskite LED. This could be attributed to the generally lower densities of states in organic semiconductors leading to smaller carrier concentrations in OLEDs[52].

Using Eq. (3) it is possible to further understand how the emissive material properties and the device design influence the apparent threshold voltages for EL (Fig. 5). An interesting observation is that higher driving voltages are required to generate the same photon flux for emissive materials with larger $E_g$ (Fig. 5a). Indeed, this offers an explanation for why the apparent threshold voltages are generally higher for wider-bandgap LEDs. Similarly, higher series resistance tends to increase the apparent

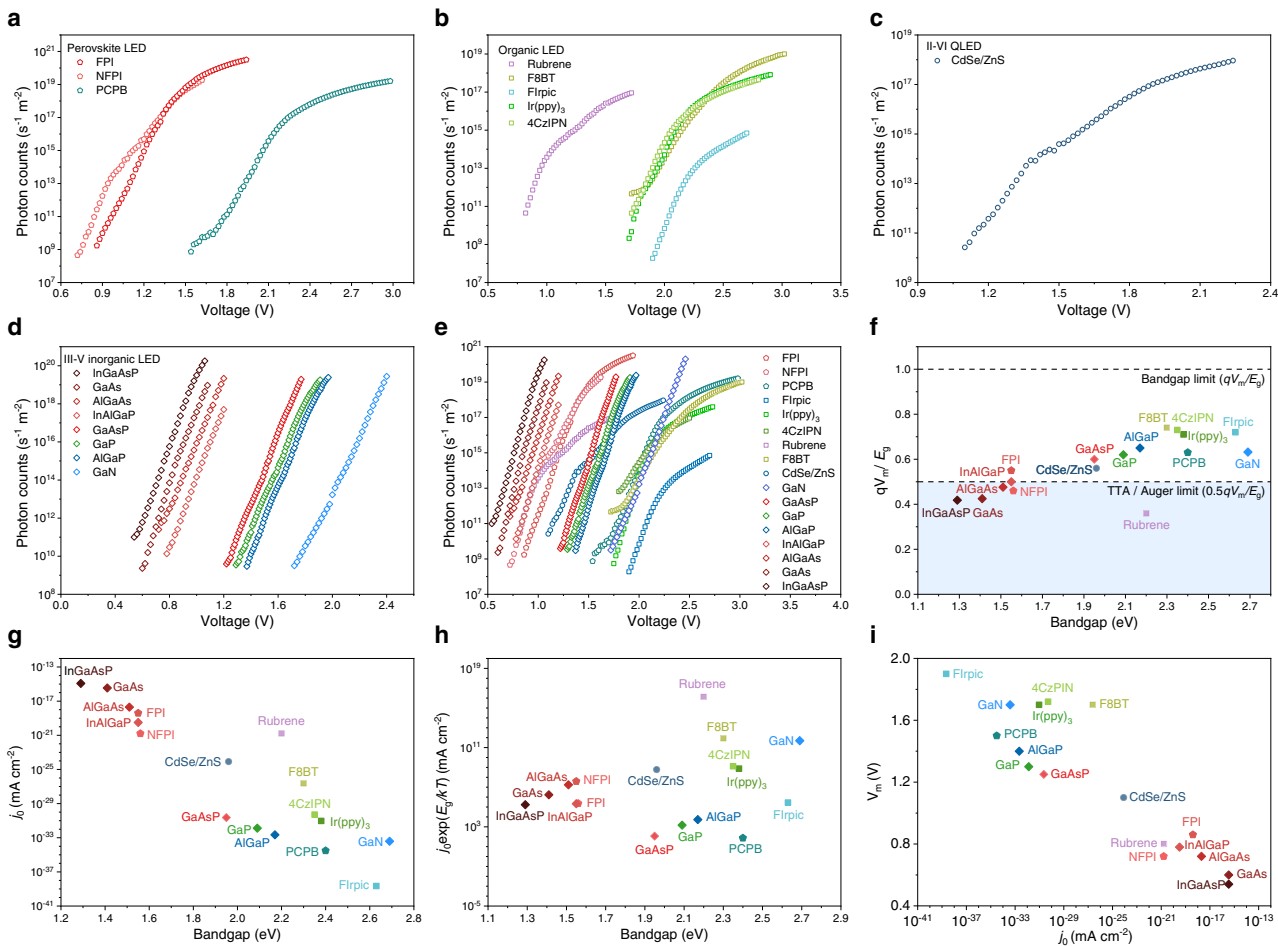

**Fig. 3 EL intensity-voltage characteristics at near- and sub-bandgap voltages for different LEDs. a** Perovskite LEDs based on FPI, NFPI and PCPB. **b** Organic LEDs based on Ir(ppy)$_3$, FIrpic, 4CzIPN, rubrene and F8BT. **c** II–VI QLED based on CdSe/ZnS QDs. **d** III–V inorganic LEDs based on InGaAsP, GaAs, AlGaAs, InAlGaP, GaAsP, GaP, AlGaP and GaN. **e** Collection of EL intensity-voltage curves for different classes of LEDs in the same panel. **f** Measured $qV_m/E_g$ of different classes of LEDs. The shaded area denotes the region where the measured $qV_m/E_g$ falls below the limits set by TTA or Auger processes. **g** Dark saturation current densities ($j_O$) of different classes of LEDs. **h** $j_O\exp(E_g/kT)$ of different classes of LEDs. **i** $V_m$ versu $j_O$ for different classes of LEDs. $V_m$ is the measured minimum voltage for detectable EL.

**Table 1 Measured minimum operating voltages of different LEDs studied in this work.**

| Device type | Emissive material | EL peak wavelength (nm) | Bandgap ($E_g$) (eV) | Measured minimum voltage ($V_m$) (V) | $\Delta E$ (eV) | $qV_m/E_g$ |
|---|---|---|---|---|---|---|
| Perovskite LED | FAPbI$_3$ (FPI) perovskite | 800 | 1.55 | 0.86 | 0.69 | 55% |
| Perovskite LED | NFPI perovskite | 790 | 1.56 | 0.72 | 0.84 | 46% |
| Perovskite LED | PCPB perovskite | 515 | 2.40 | 1.5 | 0.9 | 63% |
| Phosphorescent OLED | Ir(ppy)$_3$ | 521 | 2.38 | 1.7 | 0.68 | 71% |
| Phosphorescent OLED | FIrpic | 473 | 2.63 | 1.9 | 0.73 | 72% |
| TADF OLED | 4CzIPN | 528 | 2.35 | 1.72 | 0.63 | 73% |
| Polymer OLED | F8BT | 538 | 2.30 | 1.7 | 0.6 | 74% |
| Fluorescent small-molecule OLED | Rubrene | 563 | 2.20 | 0.8 | 1.4 | 36% |
| II–VI QLED | CdSe/ZnS | 631 | 1.96 | 1.1 | 0.86 | 56% |
| III–V inorganic LED | GaN | 461 | 2.69 | 1.7 | 1.00 | 63% |
| III–V inorganic LED | GaAsP | 570 | 2.17 | 1.25 | 0.92 | 60% |
| III–V inorganic LED | GaP | 593 | 2.09 | 1.3 | 0.79 | 62% |
| III–V inorganic LED | AlGaP | 633 | 1.95 | 1.4 | 0.55 | 65% |
| III–V inorganic LED | InAlGaP | 800 | 1.55 | 0.78 | 0.77 | 50% |
| III–V inorganic LED | AlGaAs | 820 | 1.51 | 0.72 | 0.79 | 48% |
| III–V inorganic LED | GaAs | 880 | 1.40 | 0.60 | 0.8 | 43% |
| III–V inorganic LED | InGaAsP | 960 | 1.29 | 0.54 | 0.75 | 42% |

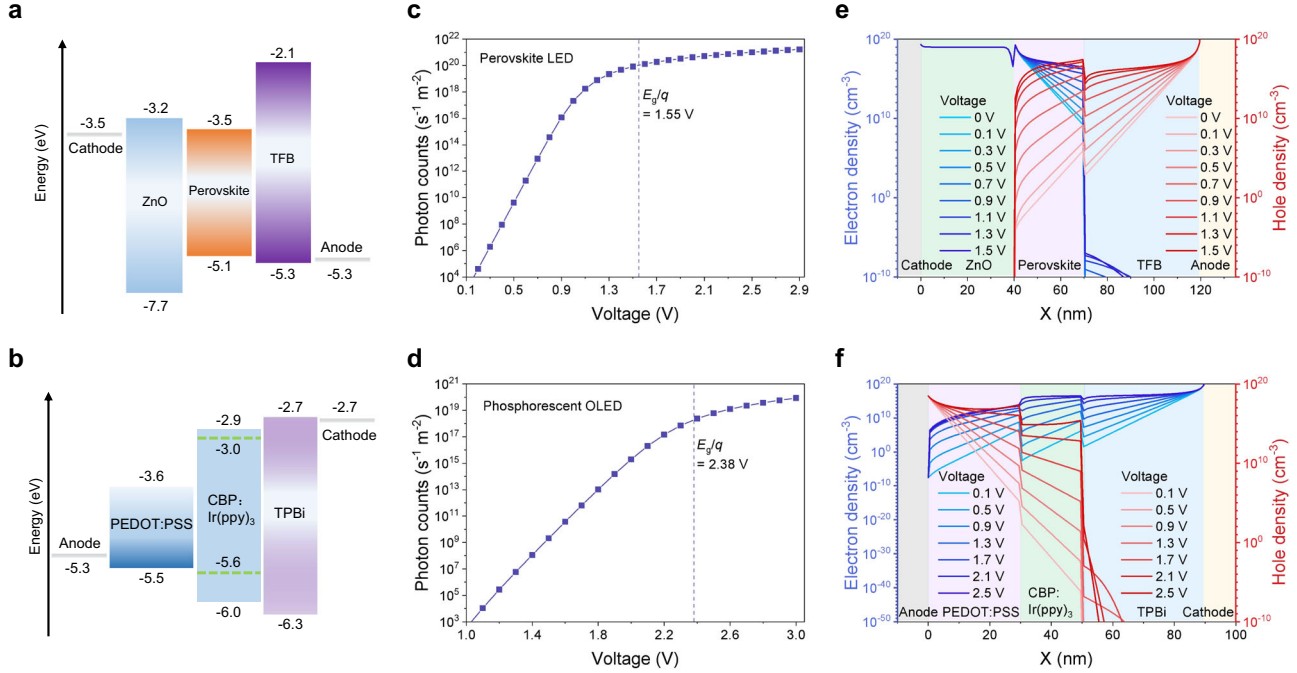

**Fig. 4 Device simulations of lead-iodide perovskite LED and Ir(ppy)₃ phosphorescent OLED using Setfos. a** Energy level diagram of a lead-iodide perovskite LED. **b** Energy level diagram of a Ir(ppy)₃ OLED. **c** Simulated EL intensity-voltage characteristics of a lead-iodide perovskite LED. **d** Simulated EL intensity-voltage characteristics of a Ir(ppy)₃ OLED. **e** Simulated electron and hole density distributions in the perovskite LED under different voltages. The shaded regions in grey, light green, pink, light blue and yellow correspond to cathode, ZnO, perovskite, TFB and anode, respectively. **f** Simulated electron and hole density distributions in the Ir(ppy)₃ OLED under different voltages. The shaded regions in grey, pink, light green, light blue and yellow correspond to anode, PEDOT:PSS, CBP:Ir(ppy)₃, TPBi and cathode, respectively.

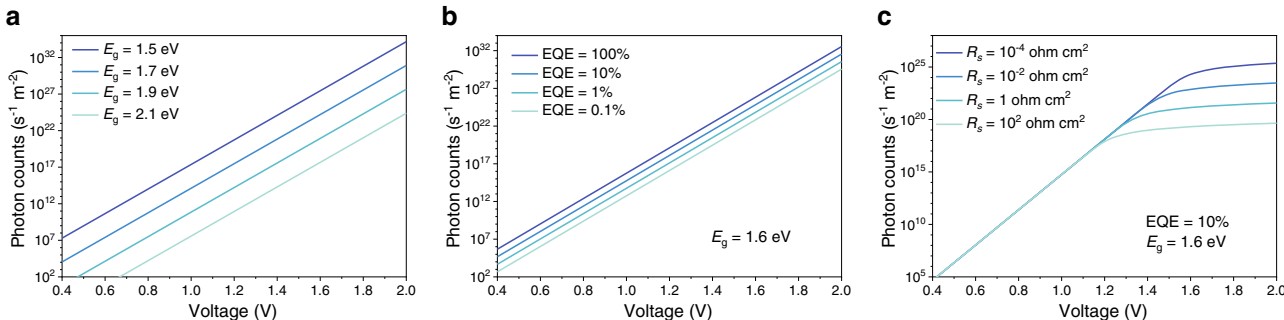

**Fig. 5 EL intensity-voltage curves generated by a simple LED model. a** Effect of bandgap. Ideal diodes with zero series resistance and an EQE of unity are assumed. **b** Effect of EQE. A bandgap of 1.6 eV with zero series resistance is assumed. **c** Effect of series resistance. A bandgap of 1.6 eV and an EQE of 10% are assumed.

threshold voltages especially under larger bias, when the fractional potential drop on the active material becomes smaller. Low-voltage operation is improved with higher EQE and with reduced series resistance (Fig. 5b, c). Using perovskite LEDs as examples, both FPI and PCPB devices showed lower apparent threshold voltages with higher EQEs (Supplementary Fig. 11a, b). In principle, series resistance consists of both bulk resistance (related to the resistivity and thickness of the layers) and contact resistance (which includes the influence of charge-injection barriers). The effects of series resistance were tested in NFPI perovskite LEDs by reducing the thickness of the hole-transport layers (HTLs), poly(N,N′-bis-4-butylphenyl-N,N′-bisphenyl)benzidine (poly-TPD). The apparent threshold voltages can be lowered from 2.4 V to 1.4 V (Supplementary Fig. 11c). Similarly, for the PCPB perovskite LEDs, by replacing the electron-transport layers (ETLs) with (1,3,5-triazine-2,4,6-triyl)tris(benzene-3,1-diyl)tris(diphenylphosphine oxide) (PO-T2T, $\mu_e$ ~4.4 × 10⁻³ cm²

V⁻¹ s⁻¹), an electron-transport material with much higher electron mobility than commonly used electron-transport materials such as bathophenanthroline (Bphen, $\mu_e$ ~5.2 × 10⁻⁴ cm² V⁻¹ s⁻¹) and 1,3,5-tris(1-phenyl-1H-benzimidazol-2-yl)benzene (TPBi, $\mu_e$ ~3.3 × 10⁻⁵ cm² V⁻¹ s⁻¹), the apparent threshold voltages could be markedly reduced from 2.8 V to 1.9 V (Supplementary Fig. 11d).

As discussed, raising the barrier height to charge injection may contribute to the series resistance of LEDs through contact resistance, potentially raising the operating voltages for the same output photon fluxes. To exemplify this effect, FPI perovskite LEDs with different HTLs (Supplementary Fig. 11e) were tested. The apparent threshold voltages are generally higher for LEDs based on HTLs with higher hole-injection barriers (Supplementary Fig. 11f). To further clarify the issue of charge injection, using Setfos we constructed a perovskite LED model with a variable charge-injection barrier at the hole-transport layer/anode

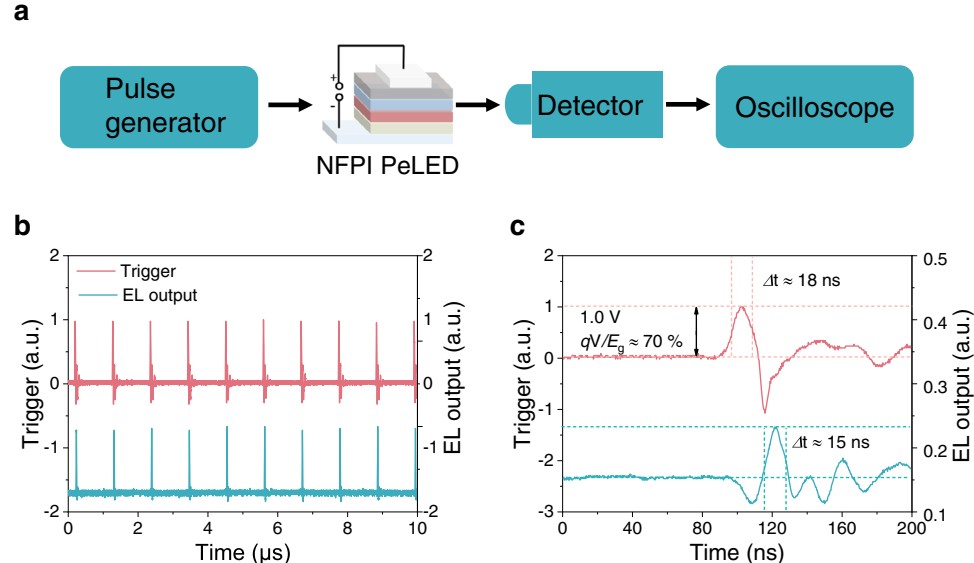

**Fig. 6 Optical pulse generation from a perovskite LED under ultralow voltages. a** Schematic diagram of a simple optical communications setup featuring an NFPI perovskite LED. **b** Characteristics of the voltage triggers from the pulse generator at a peak voltage of 1.0 V ($0.65E_g/q$) at 1 MHz and a sampling rate of 10 μs/div, and the corresponding optical pulses (EL output) from the perovskite LEDs. **c** One of input voltage triggers from the pulse generator at a sampling rate of 40 ns/div, and the corresponding EL output from the perovskite LEDs.

interface (Supplementary Fig. 12). As the barrier to hole injection increases, significant reductions of photon fluxes are observed for higher driving voltages (Supplementary Fig. 12b). The reduction in photon flux is directly related with the reduced hole densities for devices with higher hole-injection barriers (Supplementary Fig. 12c). The effect of charge-injection barrier becomes less pronounced in the low-voltage regime. These effects are consistent with the LED model (Eqs. (1)–(3)) considering charge-injection barriers as contributors to the series resistance.

We discuss below the origin of the ultralow-voltage EL phenomenon, consistent with the close relation between EL and the diode law on which the aforementioned analyses are based. Under zero bias, thermal distribution of band-edge carriers is in balance with carrier recombination (leading to $j_0$). At this point, EL does not occur as the net external current is zero due to the balance between the drift current (caused by built-in fields from band bending or interfacial dipoles) and the diffusion current (from the gradient of carrier populations) (exemplified using a generic heterojunction diode, see Supplementary Fig. 13). When a small non-zero forward bias is applied, net external current arises from the reduced drift current due to weakened built-in fields. While the band-edge carrier population due to ambient temperature remains the same, carrier recombination including radiative and non-radiative processes increases due to the injection of external charges. The radiative component of carrier recombination gives rise to EL (Supplementary Fig. 14). It is important to note that EL at sub-bandgap voltages does not violate the energy conservation principle, as the energy for photon emission is readily supplied by the recombination of charge carriers near the band edges whose distribution is governed by the Fermi-Dirac function perturbed by a small bias (Supplementary Fig. 14b). This can be supported by our temperature-dependent measurements using an FPI perovskite LED as an example. The EL intensity-temperature characteristics agree satisfactorily with the Fermi-Dirac distribution model (Supplementary Fig. 15). Device simulations using Setfos yield similar results (Supplementary Fig. 16). Compared with previously proposed mechanisms mostly for particular classes of LEDs[22–29,31–34], the present explanation is more fundamental and is applicable to a broad

range of LEDs. It also explains ultralow-voltage EL (with $V_m$ near or less than $0.5E_g/q$) that cannot be addressed by former studies.

A useful outcome of the low-voltage operation of LEDs is that these devices may be more versatile than conventional expectations. To provide an example of how this may show benefits in practical applications, we employed our NFPI perovskite LEDs in a simple optical transmitter setup (Fig. 6). With the application of sub-bandgap driving voltage of 1 V ($0.65E_g/q$), we were able to generate 1/0 (on/off) signals with a signal-to-noise ratio of 20 dB (Fig. 6b). This result is in clear contrast to former studies, where optical signal transmission based on PeLEDs[53] and CMOS-integrated LEDs[54] used bias voltages of 2.5–3 V ($\sim 2E_g/q$). The corresponding energy consumption for our LEDs to produce an optical pulse is as low as 140 pJ per bit for input frequencies ranging from 100 Hz to 1 MHz. The output pulse width is ~15 ns for an input pulse width of ~18 ns under the frequency range tested (Fig. 6b, c and Supplementary Table 5). Further reductions in energy consumption and pulse width may be possible by using a pulse generator with a smaller minimum pulse width. Remarkably, the voltage needed for optical data transmission (1 V) is lower than the silicon bandgap (1.12 eV) divided by the elementary charge. As commonly used silicon integrated circuits use 1 V chip supply voltage[55–57], LEDs operating at 1 V can be directly integrated into these circuits using the same voltage supply without the need for additional circuit elements, offering the possibility of delivering information wirelessly in optical coupling systems free from the influence of electromagnetic interference. Since it is theoretically possible to generate photons at voltages approaching zero, our results offer prospects for integrating these LEDs with low-voltage circuits for efficient optoelectronic operations, showing great potential in logic and communications applications[58,59].

## Discussion

In summary, we have demonstrated, through high-sensitivity photon detection experiments, that the voltage for EL operation could reach values below 50% of the semiconductor bandgaps, exhibiting a large apparent energy gain of 0.6–1.4 eV per emitted

photon. EL emission under ultralow voltages is a universal phenomenon across a broad range of LEDs based on perovskite, organic, II–VI chalcogenide quantum-dot and III–V semiconductors. Importantly, for 17 types of LEDs with very different modes of charge injection and recombination (e.g., dark saturation current densities ranging from $\sim10^{-39}$ to $\sim10^{-15}$ mA cm$^{-2}$), their low-voltage EL emission tracks the carrier recombination governed by the conventional diode law. Together, the experimental observation of ultralow-voltage operation of many classes of LEDs and its consistency with the diode model and device simulations, support our hypothesis that the ultralow-voltage EL arises from a universal origin – the radiative recombination of non-thermal-equilibrium band-edge carriers whose populations are determined by the Fermi-Dirac function perturbed by a small external bias. Our experiments and modelling have clarified how apparent threshold voltages can be minimised. We have demonstrated as a proof-of-concept that perovskite LEDs can transmit optical data effectively to a silicon detector at voltages below the silicon bandgap, offering prospects for data transmission at low costs. Besides establishing a unified mechanim for sub-bandgap-voltage EL, these results may lead to the underexplored territory of ultralow-voltage LEDs for communications, computational and energy applications.

## Methods

**Materials**. Poly (9, 9-dioctylfluorene-co-N-(4-(3-methylpropyl)) diphenylamine) (TFB, average molecular weight, ~50,000 g mol$^{-1}$) was purchased from Luminescence Technology Corp. Poly [N, N′-bis(4-butylphenyl)-N,N′-bis(phenyl)-benzidine] (Poly-TPD, average molecular weight, ~50,000 g mol$^{-1}$) were purchased from American Dye Source. Colloidal CdSe/ZnS core-shell red QDs and nanoparticle ZnO (30 mg ml$^{-1}$) were purchased from Guangdong Poly OptoElectronics Co., Ltd. Chlorobenzene (extra dry, 99.8%), octane (extra dry, >99%), ethanol (extra dry, 99.5%), N, N-dimethylformamide (DMF, 99.5%), Dimethyl sulfoxide (DMSO, HPLC grade) and ethyl acetate (HPLC grade) were purchased from J&K Chemical Ltd. PEDOT:PSS solution, 1-naphthylmethylamine iodide (NMAI, 99.9%), formamidinium iodide (FAI, 99.9%), PO-T2T (99.99%), MoO$_3$ (99.9%), 2-phenylethylammonium bromide (PEABr, 99.99%), poly[bis(4-phenyl)(2,4,6-trimethylphenyl)amine] (PTAA, average molecular weight, ~20,000 g mol$^{-1}$), tris(4-carbazoyl-9-ylphenyl)amine (TCTA, 99%) and BCP (99.99%) was purchased from Xi'an Polymer Light Technology Corp. Tris(2-phenylpyridine)iridium(III) (Ir(ppy)$_3$, 99%), bis[2-(4,6-difluorophenyl)pyridinato-C2,N](picolinato)iridium (FIrpic, 99%), 1,2,3,5-tetrakis(carbazol-9-yl)-4,6-dicyanobenzene (4CzIPN, 99%), poly(9,9-dioctylfluorene-alt-benzothiadiazole) (F8BT, 99%), caesium bromide (CsBr, 99.99%), lead bromide (PbBr$_2$, 99.999%), LiF (99.99%), C$_{60}$ (99.99%), 1,4,7,10,13,16-hexaoxacyclooctadecane (crown, 99%), high-purity (99.99%) rubrene, 1,1-bis[(di-4-tolylamino)phenyl]cyclohexane (TAPC, 97%) and N,N′-bis(3-methylphenyl)-N,N′-diphenylbenzidine (TPD, 99%) were purchased from Sigma-Aldrich. All materials were used as received without further purification.

**Fabrication of FPI perovskite LEDs**. The perovskite precursor solution of FAI, PbI$_2$ and 5AVA, with a molar ratio of 2:1:0.4 in DMF, was prepared to form FAPbI$_3$ perovskite (abbreviated as FPI). The molar concentration of PbI$_2$ was 0.07 M.

The device structure of FPI perovskite LEDs was ITO (135 nm)/PEIE-coated ZnO (20 nm)/FAPbI$_3$ (20 nm)/PTAA (35 nm)/MoO$_3$ (10 nm)/Au (100 nm). Solutions of ZnO nanocrystals were spin-coated onto the ITO-coated glass substrates at 5000 rpm for 60 s and annealed in air at 150 °C for 10 min. The substrates were transferred into a N$_2$ glovebox. Next, PEIE solution was spin-coated onto the ZnO surface at 5000 rpm for 60 s followed by annealing at 100 °C for 10 min. The perovskite films were prepared by spin-coating the precursor solution onto the PEIE-treated ZnO films at 5000 rpm for 90 s, followed by annealing at 100 °C for 15 min. PTAA in chlorobenzene (12 mg mL$^{-1}$) was spin-coated at 4000 rpm for 60 s. Finally, the MoO$_3$/Au electrodes were deposited using a thermal evaporation system through a shadow mask under a base pressure of $4 \times 10^{-4}$ Pa. The device area was 5.25 mm$^{-2}$ as defined by the overlapping area of the ITO films and top electrodes. All devices were encapsulated with UV epoxy (NOA81, Thorlabs)/cover glass before subsequent measurements.

**Fabrication of NFPI perovskite LEDs**. The perovskite precursor solution of 1-naphthylmethylamine iodide (NMAI), formamidinium iodide (FAI) and PbI$_2$ with a molar ratio of 2:1.8:2 dissolved in N, N-dimethylformamide (DMF) was prepared to form perovskite emissive layers with a composition of NMA$_2$FA$_{n-1}$Pb$_n$I$_{3n+1}$ (abbreviated as NFPI). The molar concentration for PbI$_2$ was 0.08 M.

The device structure of NFPI PeLEDs was ITO (135 nm)/PEIE-coated ZnO (20 nm)/NFPI (20 nm)/Poly-TPD (35 nm)/MoO$_3$ (10 nm)/Ag (100 nm). Solutions of ZnO nanocrystals were spin-coated onto the ITO-coated glass substrates at 5000 rpm for 60 s and annealed in air at 150 °C for 10 min. The substrates were transferred into a N$_2$ glovebox. Next, PEIE solution was spin-coated onto the ZnO surface at a speed of 5000 rpm for 60 s. The perovskite films were prepared by spin-coating the precursor solution onto the PEIE-treated ZnO films at 5000 rpm for 60 s, followed by annealing at 100 °C for 10 min. Poly-TPD in chlorobenzene (12 mg mL$^{-1}$) was spin-coated at 4000 rpm for 60 s. Finally, the MoO$_3$/Ag electrodes were deposited using a thermal evaporation system through a shadow mask under a base pressure of $4 \times 10^{-4}$ Pa. The device area was 5.25 mm$^{-2}$ as defined by the overlapping area of the ITO films and top electrodes. All the devices were encapsulated with UV epoxy (NOA81, Thorlabs)/cover glass before subsequent measurements.

**Fabrication of PCPB perovskite LEDs**. The perovskite precursor solution (molar ratio 5:5:2) was prepared by dissolving 110 mg lead bromide (PbBr$_2$), 64 mg caesium bromide (CsBr), and 24 mg 2-phenylethylammonium bromide (PEABr) in 1 mL dimethylsulfoxide (DMSO) and stirring overnight at room temperature.The molecular additive used was 1,4,7,10,13,16-hexaoxacyclooctadecane (crown)[60]. A quasi-2D/3D perovskite composition of PEA$_2$Cs$_{n-1}$Pb$_n$Br$_{3n+1}$ (abbreviated as PCPB) was expected to form.

The device structure of PCPB PeLEDs was ITO (185 nm)/TFB (30 nm)/LiF (1.3 nm)/PCPB (35 nm)/PO-T2T (15 nm)/LiF (0.8 nm)/Al (120 nm). Pre-patterned ITO substrates (15 ohms/square) were cleaned using ultra-sonication in acetone and isopropanol for 15 min, respectively, and then dried with a nitrogen blow gun, after which the substrates were treated under UV-Ozone for 60 min. The ITO substrates were then transferred to a nitrogen-filled glovebox. TFB was spun-coated from chlorobenzene solution (3 mg mL$^{-1}$) at 4000 rpm and was annealed at 100 °C for 10 min. 2 nm of LiF was then evaporated at a pressure of $4 \times 10^{-4}$ Pa. Subsequently, the perovskite was spin-coated from the precursor solution at 5000 rpm to form a ~35 nm layer. Finally, PO-T2T (15 nm), LiF (0.8 nm) and Al (120 nm) were sequentially evaporated through a shadow mask under a base pressure of $4 \times 10^{-4}$ Pa. The device area was 5.25 mm$^{-2}$ as defined by the overlapping area of the ITO films and top electrodes. All the devices were encapsulated with UV epoxy (NOA81, Thorlabs)/cover glass before subsequent measurements. The deposition rate for thermal evaporation was calibrated and was kept at around 2 Å s$^{-1}$ during the evaporation process for materials except LiF, for which an evaporation rate of around 0.1 Å s$^{-1}$ was used.

**Fabrication of Ir(ppy)$_3$, FIrpic and 4CzIPN OLEDs**. Molecular emitters (Ir(ppy)$_3$, FIrpic or 4CzIPN) were dissolved in DMF at a concentration of 5 mg mL$^{-1}$. PVK was dissolved in DMF at a concentration of 10 mg mL$^{-1}$. The emitter and PVK solutions were premixed to achieve the desired emitter concentrations. The overall concentration of the emissive layer (EML) solution was 7 mg mL$^{-1}$.

The device structure of the OLEDs was ITO (185 nm)/PEDOT:PSS (40 nm)/EML (35 nm)/PO-T2T (10 nm)/LiF (0.8 nm)/Al (120 nm). Pre-patterned ITO substrates (15 ohms/square) were cleaned using ultra-sonication in acetone and isopropanol for 15 min, respectively, and then dried with a nitrogen blow gun, after which the substrates were treated under UV-Ozone for 60 min. PEDOT:PSS was spin-coated on the ITO substrate at 4000 rpm for 60 s, followed by thermal annealing at 150 °C for 20 min. The thickness of the PEDOTS:PSS layer was around 40 nm. The EML was deposited by spin-coating from solution at 2000 rpm for 60 s, followed by annealing at 90 °C for 10 min, resulting in a film thickness of 50 nm. Finally, PO-T2T (10 nm), LiF (0.8 nm) and Al (120 nm) were sequentially evaporated through a shadow mask under a base pressure of $4 \times 10^{-4}$ Pa. The device area was 5.25 mm$^{-2}$ as defined by the overlapping area of the ITO films and top electrodes. All the devices were encapsulated with UV epoxy (NOA81, Thorlabs)/cover glass before subsequent measurements. The deposition rate for thermal evaporation was calibrated and was kept at around 2 Å s$^{-1}$ during the evaporation process for materials except LiF, for which an evaporation rate of around 0.1 Å s$^{-1}$ was used.

**Fabrication of rubrene OLEDs**. The device structure of rubrene-based OLEDs was ITO (185 nm)/PEDOT:PSS (30 nm)/rubrene (35 nm)/C$_{60}$ (25 nm)/BCP (6 nm)/Ag (120 nm). Organic materials were used as purchased without further purification. PEDOT:PSS was spin-coated on the substrate at 4000 rpm for 60 s, followed by annealing at 150 °C for 20 min. The thickness of the PEDOTS:PSS layer was around 40 nm. The PEDOT:PSS-coated ITO substrates were then transferred into the thermal evaporation system. A 35-nm thin layer of rubrene and a 25-nm thin layer of C$_{60}$ were deposited at a constant deposition rate of 0.5 Å s$^{-1}$. The substrate temperature was maintained at 80 °C during deposition. Further deposition was done at room temperature. A 6-nm thin layer of BCP was deposited prior to the deposition of the top electrode. Devices were completed by evaporation of a 120-nm thin layer of Ag. Metal deposition was achieved through a shadow mask. The device area was 5.25 mm$^{-2}$ as defined by the overlapping area of the ITO films and top electrodes. All depositions were performed under a base pressure lower than $4 \times 10^{-4}$ Pa. The devices were encapsulated with UV epoxy (NOA81, Thorlabs)/cover glass before subsequent measurements.

**Fabrication of polymer OLEDs based on F8BT**. The device structure of F8BT-based polymer OLEDs was ITO (185 nm)/PEDOT:PSS (30 nm)/F8BT (100 nm)/Ca (3.5 nm)/Al (120 nm). The PEDOT:PSS was spin-coated on the substrate at 7000 rpm for 60 s, followed by thermal annealing at 150 °C for 20 min. The thickness of the PEDOTS:PSS layer was around 30 nm. F8BT was deposited by spin-coating from solution (14 mg mL$^{-1}$ in chlorobenzene) at 5000 rpm, and annealed at 160 °C for 20 min, resulting in a film thickness of 50 nm. A 3.5 nm thin layer of Ca and 120 nm layer of Al were deposited by a thermal evaporation system under a base pressure lower than $4 \times 10^{-4}$ Pa. Metal deposition was achieved through a shadow mask. The device area was 5.25 mm$^{-2}$ as defined by the overlapping area of the ITO films and top electrodes. All the devices were encapsulated with UV epoxy (NOA81, Thorlabs)/cover glass before subsequent measurements.

**Fabrication of II–VI QLEDs based on CdSe/ZnS QDs**. The device structure of the II–VI QLEDs was ITO (185 nm)/PEDOT:PSS (40 nm)/TFB (25 nm)/QD (25 nm)/ZnO (65 nm)/Ag (120 nm). The PEDOT:PSS layer was deposited by a two-step spin-coating process at 1000 rpm for 10 s and 4000 rpm for 50 s, followed by annealing at 150 °C for 20 min. The PEDOT:PSS-coated substrates were transferred into a nitrogen-filled glovebox (O$_2$ < 1 ppm, H$_2$O < 1 ppm) for subsequent processes. TFB was spin-coated from solution (in chlorobenzene, 12 mg mL$^{-1}$) at 2000 rpm for 60 s and baked at 150 °C for 20 min. CdSe/ZnS QD solution (in octane, ~15 mg mL$^{-1}$) and ZnO nanocrystals (in ethanol, ~30 mg mL$^{-1}$) were sequentially spin-coated onto the substrates at 2000 rpm for 60 s. Next, Ag electrodes (120 nm) were deposited by a thermal evaporation system under a base pressure of <$4 \times 10^{-4}$ Pa. The deposition of electrodes was achieved through a shadow mask. The active area of each device was 5.25 mm$^{-2}$ as defined by the overlapping area of the ITO films and top electrodes. The devices were encapsulated with UV epoxy (NOA81, Thorlabs)/cover glass before subsequent measurements.

**Characterization of LED performance**. Current density-voltage (J–V) characteristics were measured using a Keithley 2400 source-meter unit. The luminance and EQE data were obtained using an Everfine OLED-200 commercial LED performance analysis system. The EQE measurement setup was cross-calibrated against a standard integrating sphere coupled with an Ocean Optics QE-Pro spectrometer, and with a silicon detector. The photon flux and EL spectra were measured simultaneously using a charge-coupled device centred over the light-emitting pixel. The luminance (in cd m$^{-2}$) and radiance (in W sr$^{-1}$ m$^{-2}$) of the devices were calculated based on the angular distribution functions of the LED emission and the known spectral response of the charge-coupled device. This standard setup can measure EL reliably beyond a minimum photon flux of ~$1.4 \times 10^{15}$ s$^{-1}$ m$^{-2}$ sr$^{-1}$, which corresponds to a minimum detectable photon flux of ~$10^{16}$ s$^{-1}$ m$^{-2}$ for the LED devices. Additional EL spectra of the devices driven under near- and sub-bandgap voltages were collected by a fibre-coupled focus lens and measured using a QE Pro spectrometer (Ocean Optics).

**High-sensitivity photon detection experiments**. The measurement setup for the high-sensitivity photon detection experiments is illustrated in Supplementary Fig. 4a. Photons emitted from different classes of LEDs under near-and sub-bandgap voltages were detected by a Si-based single-photon avalanche photodiode (APD). The APD converts the photons from the LEDs into photocurrent, which is amplified by an amplifier. The photocurrent forms sharp pulses through a pulse shaper. These pulses are transmitted effectively through a BNC wire with low signal distortion. The controller converts the pulses to photon counts before transmitting data to the computer. It has an instrumental response time of ~0.2 ns. For each measurement, the minimum counts on the APD is on the order of 1000 s$^{-1}$, corresponding to a minimum detectable photon flux of ~$10^9$ s$^{-1}$ m$^{-2}$ for the LED devices.

The raw EL intensity data collected using the high-sensitivity photon detection system are shown in Supplementary Fig. 4b–l. Due to the finite collection efficiency and the intrinsic saturation characteristics of the APD, the raw data counts do not follow a linear relationship with the actual photon counts from the EL of the LEDs. In this work, we used the EL intensity-voltage data obtained from the commercial LED measurement system to calibrate the photon count-voltage response of the high-sensitivity system by driving the same LED under an identical voltage range. To extend the measurement range and to avoid saturation of the APD, EL from the LEDs was attenuated by a neutral density filter before entering the APD. The transfer function g(x) for the calibration of the high-sensitivity setup can be expressed by

$$APD\,counts = g(Photon\,counts \times w)$$

where w is the attenuation factor set by the neutral density filter and the collection efficiency of the optics. It is possible to calculate the actual photon counts from the raw data collected from the APD, according to

$$Photon\,counts = 1/w \times g^{-1}(APD\,counts)$$

where $g^{-1}(x)$ is the inverse function of g(x).

**LED device simulations**. In addition to the LED modelling and EL intensity-voltage data fitting presented in this work, we carried out device simulations for a lead-iodide perovskite LED and a Ir(ppy)$_3$ OLED using a LED simulation software "Setfos"[50,51]. The preset parameters for these devices are available from the database of the device simulation package, with detailed settings and modifications presented in Supplementary Tables 3 and 4.

## Data availability

The main data supporting the findings of this study are available within the article and its Supplementary Information. Additional data are available from the corresponding author upon reasonable request.

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

## Acknowledgements

This work was supported by the National Key Research and Development Program of China (grant no. 2018YFB2200401) (D.D.), the National Natural Science Foundation of China (NSFC) (61975180 (D.D.), 62005243 (B.Z.), 22103069 (R.L.)), Kun-Peng Programme of Zhejiang Province (D.D.), the Natural Science Foundation of Zhejiang Province (LR21F050003) (B.Z.), the Fundamental Research Funds for the Central Universities (2020QNA5002) (B.Z.), Zhejiang University Education Foundation Global Partnership Fund (D.D.), and the Engineering and Physical Sciences Research Council (EPSRC) (R.H.F.). The authors acknowledge the administrative support from Minhui Yu and Yuzhen Zhao.

## Author contributions

D.D. conceived the project and planned the study. Y.L. and S.X. fabricated and characterized the LEDs. Y.L. performed the high-sensitivity EL experiments. D.L. established the diode model and derived relevant equations. D.L. and S.X. carried out data fitting and analysis. B.G., R.L., Z.R., C.Z. and B.Z. contributed to experiments and analyses. D.D., Y.L., S.X. and D.L. wrote the manuscript. R.H.F. provided suggestions for substantial revisions. All authors contributed to the work and commented on the paper.

## Competing interests

D.D., Y.L., S.X., D.L., B.Z., and B.G. are inventors on CN patent application: 202110879765.7. The remaining authors declare no competing interests.
