## [Peer Review File · Nature Communications]

Ultralow-voltage operation of light-emitting diodesREVIEWER COMMENTS

Reviewer #1 (Remarks to the Author):

In this work, Di and coworkers investigate light emission from LEDs when exciting at low bandgap. Across a wide variety of material systems, they show a surprising amount of emission well below the bandgap. They model this with a simple diode equation and demonstrate reasonable fits, and explain the results with a simple qualitative model. They finally demonstrate data transmission at 1V. This paper is potentially publishable eventually due to the interesting topic and huge amount of work by the authors, however the lack of clarity on the mechanism, as well as a few other challenges, prevent me from recommending publication at this time.

The first challenge to this work is understanding the mechanism. The diode fit is reasonable enough, and fits the data well, although there are key discussions (such as the ideality factor) omitted. However, this model does not itself explain the mechanism, as that's merely incorporated into an "EQE" factor. The proposed mechanism (Fig S9/10) is qualitative and needs to be much better understood. There are innumerable device physics models, both software and closed form, that could simulate the device to generate more clear understanding. This needs to be done, to either verify the proposed mechanism or discuss an alternative. Along those lines, although the energy conservation argument is dismissed due to the distribution of carriers, key components of the process such as charge injection are left undiscussed. The mechanism needs to be made much clearer.

Further, the authors repeatedly oversell the engineering value of this process. As an example, they describe their data transmission as "efficient" yet at the 1V bias the EQE of the devices is ~0.001%! Under no circumstances does that EQE count as efficient. The EQEs, presented in the supplementary, are swept under the rug, yet they are an important piece of the puzzle. To be clear, this paper is interesting already due to the scientific demonstration, it does not need this false engineering inflation.

I do wish to congratulate the authors for the huge amount of work done. Building such a wide variety of LEDs is difficult in the extreme, and the universal understanding generated from that is why I am potentially supportive of this paper, even with the above difficulties.

Other points:

- Could the authors normalize Figs 2/S2/S3? The most interesting curves (the lowest biases) are difficult to see.

- What is happening with the rubrene devices, in particular the EQE?

Reviewer #2 (Remarks to the Author):

Lian et al. studied a physical origin of photon emission under sub-bandgap voltage application in conventional EL devices. The authors demonstrate that a traditional principles of semiconductor diodes can explain well the experimental results for EL devices using many classes of materials. The results presented here not only address a question that has been recently a subject of debate in high IF journals, but are also useful for future developments such as single photon light sources. I recommend a publication of this paper provided that the following minor issues are addressed.

- In the highly sensitive photon detection measurements, the authors should somehow prove that the detected photon energy corresponds to the bandgap of the material. The results in Fig. 2 are inadequate as they were measured at relatively high voltages. Bandpass filters, for example, would be available.

- I understood the authors conclude that the small perturbation of Fermi-Dirac distribution due to the small current injection causes the ultralow-voltage EL. Basically I also agree with the authors' claims. To reinforce this claim, the author can employ temperature dependent measurement. It would be OK to test this for any one type of LED.

- In connection with the above question, the model suggested by the authors is based on a principle established for conventional inorganic semiconductors. Therefore it seems a little strange that the same physics holds for material classes that show different density of state, such as quantum dots and organic semiconductors. Yes, it is one of the important points that the authors found, but it should be better to provide more discussion or comment on this point to eliminate this concern.

Reviewer #3 (Remarks to the Author):

This paper by Lian et al investigates the lowest operation voltages of a series of LEDs made with inorganics, organics, QD and perovskites emission layers. They have discovered that an ultra-low on-set voltage can initiate the EL from all LEDs investigated in this study, which correspond to energies below the optical band gap. They have found the V_m to be inversely proportional to the J_0 , a measure of the dark recombination in the device. Finally, they have demonstrated the benefit of using an ultra-low voltage LED for optical communication, which is novel.

Overall, this is a very interesting and sound investigation. The low V_{on} has indeed been observed in perovskite LEDs, and this paper provides a comprehensive explanation with sound experimental support. The mechanism is solidly discussed through a simple diode equation, and is supported by modeling and experimental data. Such a sound investigation will bring significant value to the field of LEDs using different material systems. I would thus recommend its publication in a high-profile journal like Nature Comm.

One minor point worth discussing, is that the authors found V_m is lower when J_0 is higher. Here J_0 represent the dark recombination by injected carrier, and perhaps is composed with thermal recombination via trap states. Therefore, what is the role of trap states (bulk and interface). In other words, does this relation hold in present of trap? Another point worth understanding is the imbalanced charge injection. What if one carrier is faster than the other, and large J_0 could be also obtained by large recombination near the interface, and what could be expected in this case?

Response to Reviewers' Comments

Reviewer #1

In this work, Di and coworkers investigate light emission from LEDs when exciting at low bandgap. Across a wide variety of material systems, they show a surprising amount of emission well below the bandgap. They model this with a simple diode equation and demonstrate reasonable fits, and explain the results with a simple qualitative model. They finally demonstrate data transmission at 1V. This paper is potentially publishable eventually due to the interesting topic and huge amount of work by the authors, however the lack of clarity on the mechanism, as well as a few other challenges, prevent me from recommending publication at this time.

Response: We thank the reviewer for the supportive and helpful comments, which encouraged us to prepare a substantially improved paper.

The first challenge to this work is understanding the mechanism. The diode fit is reasonable enough, and fits the data well, although there are key discussions (such as the ideality factor) omitted. However, this model does not itself explain the mechanism, as that's merely incorporated into an "EQE" factor. The proposed mechanism (Fig S9/10) is qualitative and needs to be much better understood. There are innumerable device physics models, both software and closed form, that could simulate the device to generate more clear understanding. This needs to be done, to either verify the proposed mechanism or discuss an alternative. Along those lines, although the energy conservation argument is dismissed due to the distribution of carriers, key components of the process such as charge injection are left undiscussed. The mechanism needs to be made much clearer.

Response: We are grateful to the reviewer for the insightful comments. We agree that validating and clarifying the proposed mechanism are important, and have now resolved the issues raised by the reviewer as follows.

Discussion on ideality factor. We agree with the reviewer that a discussion on ideality factor is necessary. To address this point, we have now stated on page 7 (highlighted): "The ideality factors

presented in Supplementary Table 3 fall within the range between 1 and 2. Smaller n (closer to 1) suggests a reduced fraction of defect-assisted recombination, generally corresponding to higher-quality diodes with efficient charge transport and radiative recombination. This is the case for LEDs based on III-V semiconductors and PeLEDs based on FPI. The ideality factors for most of the PeLEDs, QLEDs and OLEDs tested are slightly higher, likely due to the reason that the charge transport and recombination processes are influenced by traps^{48,49}.”

Device simulations using a commercial software. To further verify our proposed mechanism for ultralow-voltage EL, we followed the reviewer’s advice and employed a widely used LED simulation software “Setfos” (as used in e.g. *Nature* 2021, 599, 594-598; *Organic Electronics* 2022, 101, 106423; *Advanced Materials* 2022, 34, 2200526) to model the emission behaviour of devices. We constructed model devices including a lead-iodide perovskite LED, and a standard phosphorescent OLED based on Ir(ppy)₃ (Supplementary Fig. 11). The simulation results (Supplementary Fig. 11 c&d) show that both types of LEDs are capable of generating EL at voltages well below the bandgaps in a fashion similar to what we have observed with our experimental setup (Fig. 3), consistent with the LED model we described. From the simulations, it can be seen that the intensities of the output photon fluxes correlate strongly with the densities of injected charges. At operating voltages significantly below the bandgap, there are appreciable levels of electron and hole populations contributing to the radiative recombination. These results are consistent with the diode law and with our proposed mechanism for sub-bandgap EL. At similar photon fluxes, the modelled Ir(ppy)₃ OLED operates at higher voltages in comparison to that of the perovskite LED. This could be attributed to the generally lower densities of states in organic semiconductors leading to smaller carrier concentrations in OLEDs. The related discussion to address this point is now presented on page 8 (highlighted).

Supplementary Figure 11 | Device simulations of lead iodide perovskite LED and Ir(ppy)₃ phosphorescent OLED using Setfos. a, Energy level diagram of a lead iodide perovskite LED. **b,** Energy level diagram of a Ir(ppy)₃ OLED. **c,** Simulated EL intensity-voltage characteristics of a lead iodide perovskite LED. **d,** Simulated EL intensity-voltage characteristics of a Ir(ppy)₃ OLED. **e,** Simulated electron and hole density distributions in the perovskite LED under different voltages. **f,** Simulated electron and hole density distributions in the Ir(ppy)₃ OLED under different voltages.

Effects of charge injection with a varying barrier height. To clarify the issue of charge injection, using Setfos we constructed a perovskite LED model with a variable charge injection barrier at the hole-transport layer/anode interface. As the barrier to hole injection increases, significant reductions of photon fluxes are observed for higher driving voltages (Supplementary Fig. 13b). The reduction in photon flux is directly related with the reduced hole densities for devices with higher hole-injection barriers (Supplementary Fig. 13c). The effect of charge-injection barrier becomes less pronounced in the low-voltage regime. These effects are consistent with the LED model in Eq. (1)-(3). As the charge-injection barrier raises the interfacial contact resistance which contributes to the series resistance (R_s) of the diodes, its effect is expected to be less pronounced in the low-voltage regime. This issue has now been discussed on page 9 (highlighted) of the revised paper.

Supplementary Figure 13 | Simulated EL intensity-voltage characteristics and carrier density distributions for lead iodide PeLEDs with variable hole-injection barriers. **a**, Energy level diagram of a lead iodide perovskite LED with a variable anode/HTL barrier ($\Delta E_{\text{barrier}}$). $\Delta E_{\text{barrier}} = |E_{\text{HOMO,TFB}} - |E_{f,\text{anode}}|$, where $E_{f,\text{anode}}$ is the Fermi level of the anode, and $E_{\text{HOMO,TFB}}$ is the highest occupied molecular orbital (HOMO) level of the TFB HTL. **b**, Effect of the anode/HTL barrier height on the EL intensity-voltage characteristics. **c**, Effect of the anode/HTL barrier height on carrier density distribution in a lead iodide perovskite LED ($E_g = 1.55$ eV) under a sub-bandgap driving voltage of 1 V.

Further discussions. Together, the experimental observation of the ultralow-voltage operation of many classes of LEDs and its consistency with the diode model and with the device simulations, support our view that the ultralow-voltage EL arises from a universal origin – the radiative recombination of non-thermal-equilibrium band-edge carriers whose populations are determined by the Fermi-Dirac function perturbed by a small external bias. For further confirmation, using Setfos we find that the simulated carrier densities of a lead iodide perovskite LED at different temperatures can be fitted satisfactorily using Fermi-Dirac statistics (Supplementary Fig. 17). Further discussions on this point are provided on pages 10 & 12 of the revised paper (highlighted).

Supplementary Figure 17 | Simulated carrier density-temperature characteristics of a lead iodide perovskite LED. The carrier densities at different temperatures are normalized to unity at the highest temperature. In these simulations, the carrier densities at the centre of the emissive layer are taken as the representative data points. **a**, Hole density-temperature

characteristics for different driving voltages. **b**, Electron density-temperature characteristics for different driving voltages. The data can be fitted satisfactorily using the Fermi-Dirac statistics. The density of electrons, $n_e \propto \int_{E_g}^{\infty} \frac{1}{1+e^{-\frac{(E-E_{fe})}{kT}}} \sqrt{E-E_g} dE$. The density of holes, $n_h \propto \int_{E_g}^{\infty} \frac{1}{1+e^{-\frac{(E-E_{fh})}{kT}}} \sqrt{E-E_g} dE$. Here, E_{fe} and E_{fh} are the quasi-Fermi levels of electrons and holes; The quasi-Fermi level splitting $E_{fe}-E_{fh}$ is governed by the driving voltage; E_g is the bandgap; k is the Boltzmann constant; T is the temperature.

Further, the authors repeatedly oversell the engineering value of this process. As an example, they describe their data transmission as “efficient” yet at the 1V bias the EQE of the devices is ~0.001%! Under no circumstances does that EQE count as efficient. The EQEs, presented in the supplementary, are swept under the rug, yet they are an important piece of the puzzle. To be clear, this paper is interesting already due to the scientific demonstration, it does not need this false engineering inflation.

Response: We thank the reviewer for the criticism on the engineering aspects, and for recognizing the scientific value of our work. We fully agree with the reviewer that the primary focus of our work lies in the scientific demonstration rather than in the engineering applications. To address this issue, we have replaced the word “efficient” with “effective” (highlighted) to avoid overclaiming of the applied aspects of the work.

I do wish to congratulate the authors for the huge amount of work done. Building such a wide variety of LEDs is difficult in the extreme, and the universal understanding generated from that is why I am potentially supportive of this paper, even with the above difficulties.

Response: We thank the reviewer for the encouraging words. We study the effect of ultralow-voltage EL using the widest range of LEDs attainable to ensure a high level of confidence for our results. We hope the reviewer would find that our paper substantially improved following the expert comments is now in a form suitable for publication in *Nature Communications*.

Other points:

- Could the authors normalize Figs 2/S2/S3? The most interesting curves (the lowest biases) are difficult to see.

Response: We appreciate the helpful suggestions. The normalized spectra are now presented in Supplementary Figs. 2 & 3. The spectra under low and high voltages are nearly identical.

- What is happening with the rubrene devices, in particular the EQE?

Response: We thank the reviewer for the comments. Efficient rubrene OLEDs are normally based on host-guest structures (e.g. *Adv. Mater.* 2017, **29**, 1605987), in which the wide-bandgap organic host raises the series resistance of the devices, increasing the apparent threshold voltages of the LEDs. In this work, the rubrene devices are prepared in a host-free configuration, with the rubrene emissive layer sandwiched between highly conductive charge-transport materials, PEDOT:PSS and C60. These rubrene OLEDs are expected to show reduced series resistance, consistent with the very low apparent threshold voltages observed. However, the light-emitting performance of the host-free rubrene layer is limited by the PEDOT:PSS and C60 layers which are exciton quenchers, leading to the significantly reduced EQEs. Nevertheless, these results strengthen the view that ultralow-voltage LED operation is a universal phenomenon regardless of material class and device efficiency.

To address this issue, we have now provided information on this point in the caption of Supplementary Fig. 1 (highlighted).

Reviewer #2

Lian et al. studied a physical origin of photon emission under sub-bandgap voltage application in conventional EL devices. The authors demonstrate that a traditional principles of semiconductor diodes can explain well the experimental results for EL devices using many classes of materials. The results presented here not only address a question that has been recently a subject of debate in high IF journals, but are also useful for future developments such as single photon light sources. I recommend a publication of this paper provided that the following minor issues are addressed.

Response: We are grateful to the reviewer for the supportive and insightful comments.

- In the highly sensitive photon detection measurements, the authors should somehow prove that the detected photon energy corresponds to the bandgap of the material. The results in Fig. 2 are inadequate as they were measured at relatively high voltages. Bandpass filters, for example, would be available.

Response: We thank the reviewer for the valuable suggestions. Following the advice, we carried out high-sensitivity photon detection measurements with and without bandpass filters for various types of LEDs (Supplementary Fig. 5). The filters were selected based on the criterion that the cut-off wavelengths are reasonably close to the materials' bandgaps. We found that the photon count-voltage curves for each class of LEDs are nearly identical for cases with and without the filters – they all exhibit ultralow voltage operation. These results support the view that the sub-bandgap photons detected arise from the recombination of band-edge carriers, rather than from sub-bandgap states.

This issue has been addressed through new experiments presented in Supplementary Fig. 5, with discussions on page 5 (highlighted).

Supplementary Figure 5 | EL measurements with and without bandpass filters. a, EL spectra of LEDs and the transmission spectra of the bandpass filters. **b,** EL intensity-voltage characteristics of FPI perovskite LEDs. **c,** EL intensity-voltage characteristics of Ir(ppy)₃ OLEDs. **d,** EL intensity-voltage characteristics of GaAsP LEDs.

- I understood the authors conclude that the small perturbation of Fermi-Dirac distribution due to the small current injection causes the ultralow-voltage EL. Basically I also agree with the authors' claims. To reinforce this claim, the author can employ temperature dependent measurement. It would be OK to test this for any one type of LED.

Response: We are grateful to the reviewer for the insightful comments. Following the advice, we have now carried out temperature-dependent measurements for the FPI perovskite LEDs as an example to further assess the validity of our claims. We found that the temperature-dependent EL intensity characteristics of the LEDs obey the Fermi-Dirac statistics (Supplementary Figure 16). We carried out additional device modelling using a LED simulation software “Setfos” (also discussed in the point below), which similarly gives a temperature-dependent relationship for the carrier densities in agreement with the Fermi-Dirac distribution model (Supplementary Fig. 17).

To address this issue, we have provided further discussions on page 10 (highlighted) of the revised manuscript, with experimental and simulation results presented in Supplementary Figs. 16&17.

Supplementary Figure 16 | EL intensity-temperature characteristics of a FPI perovskite LED. The EL intensities at different temperatures are normalized to unity at the highest temperature. The measured EL intensity (I_{EL})-temperature data can be fitted satisfactorily according to $I_{EL} \propto \int_{E_g}^{\infty} f(E) N_j dE$, where E_g is the bandgap, and $f(E) = \frac{1}{1 + e^{\frac{E - (E_{fe} - E_{fh})}{kT}}}$, which gives the distribution of carriers governed by the Fermi-Dirac function. E_{fe} and E_{fh} are the quasi-Fermi levels of electrons and holes; the quasi-Fermi level splitting $E_{fe} - E_{fh}$ is determined by the driving voltage. k is the Boltzmann constant; T is the temperature. $N_j(E) = \frac{m_j^*}{\pi^2 \hbar^3} \sqrt{E - E_g}$ is the joint density of states; m_j^* is the reduced effective mass defined by $\frac{1}{m_j^*} = \frac{1}{m_e^*} + \frac{1}{m_h^*}$, where m_e^* and m_h^* are the effective masses of electrons and holes, respectively⁶⁰.

Supplementary Figure 17 | Simulated carrier density-temperature characteristics of lead iodide perovskite LED. The carrier densities at different temperatures are normalized to unity at the highest temperature. In these simulations, the carrier densities at the centre of the emissive layer are taken as the representative data points. **a**, Hole density-temperature characteristics for different driving voltages. **b**, Electron density-temperature characteristics for different driving voltages. The data can be fitted satisfactorily using the Fermi-Dirac statistics. The density of electrons, $n_e \propto \int_{E_g}^{\infty} \frac{1}{1 + e^{\frac{E - E_{fe}}{kT}}} \sqrt{E - E_g} dE$. The density of holes, $n_h \propto \int_{E_g}^{\infty} \frac{1}{1 + e^{\frac{E - E_{fh}}{kT}}} \sqrt{E - E_g} dE$. Here, E_{fe} and E_{fh} are the quasi-Fermi

levels of electrons and holes; The quasi-Fermi level splitting $E_{je}-E_{jn}$ is determined by the driving voltage; E_g is the bandgap; k is the Boltzmann constant; T is the temperature.

- In connection with the above question, the model suggested by the authors is based on a principle established for conventional inorganic semiconductors. Therefore it seems a little strange that the same physics holds for material classes that show different density of state, such as quantum dots and organic semiconductors. Yes, it is one of the important points that the authors found, but it should be better to provide more discussion or comment on this point to eliminate this concern.

Response: We appreciate the helpful suggestions from the reviewer. To further verify our proposed mechanism for ultralow-voltage EL, we employed a widely used LED simulation software “Setfos” (as used in e.g. *Nature* 2021, 599, 594-598; *Organic Electronics* 2022, 101, 106423; *Advanced Materials* 2022, 34, 2200526) to model the emission behaviour of devices. We constructed model devices including a lead-iodide perovskite LED, and a standard phosphorescent OLED based on Ir(ppy)₃ (Supplementary Fig. 11). The simulation results (Supplementary Fig. 11 c&d) show that both types of LEDs are capable of generating EL at voltages well below the bandgap in a fashion similar to what we observed experimentally (Fig. 3), consistent with the LED model we described. From the simulations, it can be seen that the intensities of the output photon fluxes correlate strongly with the densities of the injected charges. At low operating voltages significantly below the bandgap, there are appreciable levels of electron and hole populations contributing to the radiative recombination (Supplementary Fig. 11e&f). These results are consistent with the diode law and with our proposed mechanism for sub-bandgap EL. At similar photon fluxes, the modelled Ir(ppy)₃ OLED operate at higher voltages in comparison to that of the perovskite LED. This could be attributed to the generally lower densities of states in organic semiconductors leading to smaller carrier concentrations in OLEDs.

The simulation results provide two important outcomes: (1) the results further confirm the suitability of the proposed model for LEDs operating at lower voltages (below the photon detection limits of our experimental setup), where the simulated EL-voltage characteristics continues to follow the same behaviour; (2) the results reveal that the recombination of injected carriers at ultralow (non-zero) voltages is sufficient for generating EL.

Supplementary Figure 11 | Device simulation of lead iodide perovskite LED and Ir(ppy)₃ phosphorescent OLED using Setfos. **a**, Energy level diagram of a lead iodide perovskite LED. **b**, Energy level diagram of a Ir(ppy)₃ OLED. **c**, Simulated EL intensity-voltage characteristics of a lead iodide perovskite LED. **d**, Simulated EL intensity-voltage characteristics of a Ir(ppy)₃ OLED. **e**, Simulated electron and hole density distributions in the perovskite LED under different voltages. **f**, Simulated electron and hole density distributions in the Ir(ppy)₃ OLED under different voltages.

Together, the experimental observation of the ultralow-voltage operation of many classes of LEDs and its consistency with the diode model and device simulation, support our hypothesis that the ultralow-voltage EL arises from a universal origin – the radiative recombination of non-thermal-equilibrium band-edge carriers whose populations are determined by the Fermi-Dirac function perturbed by a small external bias. It is indeed surprising that the diode law and its underlying physics are universally applicable in the ultralow-voltage regime for LEDs based on a wide range of materials systems with very different densities of states and charge-transport processes.

Discussions around this point are now provided on pages 8 & 12 (highlighted) of the revised paper, with new device modelling results presented in Supplementary Fig. 11.

Reviewer #3

This paper by Lian et al investigates the lowest operation voltages of a series of LEDs made with inorganics, organics, QD and perovskites emission layers. They have discovered that an ultra-low on-set voltage can initiate the EL from all LEDs investigated in this study, which correspond to energies below the optical band gap. They have found the V_m to be inversely proportional to the J_0 , a measure of the dark recombination in the device. Finally, they have demonstrated the benefit of using an ultra-low voltage LED for optical communication, which is novel.

Overall, this is a very interesting and sound investigation. The low V_{on} has indeed been observed in perovskite LEDs, and this paper provides a comprehensive explanation with sound experimental support. The mechanism is solidly discussed through a simple diode equation, and is supported by modeling and experimental data. Such a sound investigation will bring significant value to the field of LEDs using different material systems. I would thus recommend its publication in a high-profile journal like Nature Comm.

Response: We thank the reviewer for the encouraging and helpful comments, which allowed us to carry out new experiments and analyses to strengthen the paper.

One minor point worth discussing, is that the authors found V_m is lower when J_0 is higher. Here J_0 represent the dark recombination by injected carrier, and perhaps is composed with thermal recombination via trap states. Therefore, what is the role of trap states (bulk and interface). In other words, does this relation hold in presence of trap? Another point worth understanding is the imbalanced charge injection. What if one carrier is faster than the other, and large J_0 could be also obtained by large recombination near the interface, and what could be expected in this case?

Response: We are grateful to the reviewer for raising this interesting point. The reviewer is correct that experimentally, there is a negative correlation between V_m and j_0 across a range of material systems we investigated (Fig. 3i). This interesting correlation can be understood using Eq. (1), (2) and (S7): to generate a certain level of photon flux (e.g. that corresponds to the sensitivity limit of the experimental setup), LEDs with larger j_0 generally require smaller driving voltages. However, as the reviewer suggested, such correlation may not necessarily hold when other factors, such as trap-mediated non-

radiative recombination that affects the radiative efficiency, become dominant. In light of the reviewer's suggestions, we break down the problem into two parts, including the effects of traps and the influence of imbalanced charge injection.

The effects of traps have now been investigated by tuning the emission quantum efficiency of the emissive layer in PCPB PeLEDs using molecular additives (Supplementary Fig. 9). The PLQEs and device EQEs are used as indicators of the strength of trap-mediated non-radiative recombination (Supplementary Fig. 9a). From this set of experiments, we found that the minimum measurable voltage for EL (V_m) generally increases as j_0 increases (Supplementary Fig. 9b,c), in contrast to the V_m - j_0 correlation summarised for a range of LED classes presented in Fig. 3i. As discussed above, such difference is due to the reason that V_m is negatively correlated with the radiative efficiency (Supplementary Fig. 9d), which plays a dominant role in the case of PCPB PeLEDs.

Supplementary Figure 9 | Characteristics of PCPB perovskite LEDs with different molar fractions of molecular additives. The molecular additive used was 1,4,7,10,13,16-hexaaxacyclooctadecane (crown). The molar ratio of the crown additive versus Pb in the precursor solution was tuned from 0%-17%. **a**, PLQEs of perovskite films, and EQEs of perovskite LEDs based on the same compositions as for the PLQE experiments. **b**, EL intensity-voltage characteristics. **c**, V_m versus j_0 . **d**, V_m versus peak EQE.

The effect of imbalanced charge injection has been studied using PCPB PeLEDs with electron-transport layers (ETL) of various thicknesses (Supplementary Fig. 10). The variation in the ETL thickness is expected to adjust the degree of charge injection balance, leading to changes in device EQEs (Supplementary Fig. 10a). In this set of experiments, we find that V_m does not show a clear correlation with j_0 (Supplementary Fig. 10b&c). Instead, we find that V_m reduces as peak EQE increases (Supplementary Fig. 10d), consistent with the view that improved radiative efficiency and charge injection balance are beneficial in reducing the apparent threshold voltages.

Supplementary Figure 10 | Characteristics of PCPB perovskite LEDs with electron-transport layer (ETL) thickness variation. The ETL used was 2,4,6-tris[3-(diphenylphosphinyl)phenyl]-1,3,5-triazineis (PO-T2T). **a**, EQE-current density curves. **b**, EL intensity-voltage characteristics. **c**, V_m versus j_0 . **d**, V_m versus peak EQE.

Overall, among the many factors affecting j_0 , the emissive material's bandgap E_g , on which j_0 is exponentially dependent, plays a significant role (as evidenced in Fig. 3g). As such, LEDs based on materials with smaller E_g normally have larger j_0 (presuming other factors such as the trap states have smaller influence), leading to the normally negative correlation between j_0 and V_m . However, larger j_0 may also arise from a higher density of defect states, particularly for LEDs based on the same or similar emissive materials, in which case the same V_m - j_0 correlation may no longer hold, as V_m is negatively

correlated with the emission efficiency (see e.g. Supplementary Fig. 9). In a related case of imbalanced charge injection where one type of charge carriers (either electrons or holes) dominate over the other, V_m may not show a negative correlation with j_0 (see e.g. Supplementary Fig. 10). This is because j_0 may partly originate from trap-assisted non-radiative recombination (in the bulk or at the charge-transport interfaces), which cannot contribute to the EL.

This issue has now been discussed on page 7-8 (highlighted) of the revised manuscript, with new experiments presented in Supplementary Fig. 9 & 10.

REVIEWERS' COMMENTS

Reviewer #1 (Remarks to the Author):

I congratulate the authors on excellent experiments, simulations, and clarifications to the manuscript. I am happy to recommend publication.

Reviewer #2 (Remarks to the Author):

The authors have properly addressed all issues raised by three reviewers. They newly provided excellent results of additional experiment and calculation, demonstrating solid evidences of the EL mechanism they claim. As a result, the academic importance and technical excellence of this manuscript are even more striking. I recommend the publication of this work in Nature Communications in the present form.

Reviewer #3 (Remarks to the Author):

The authors have adequately addressed my comments; I support its publication.

Response to Reviewers' Comments:

Reviewer #1

I congratulate the authors on excellent experiments, simulations, and clarifications to the manuscript. I am happy to recommend publication.

Reviewer #2

The authors have properly addressed all issues raised by three reviewers. They newly provided excellent results of additional experiment and calculation, demonstrating solid evidences of the EL mechanism they claim. As a result, the academic importance and technical excellence of this manuscript are even more striking. I recommend the publication of this work in Nature Communications in the present form.

Reviewer #3

The authors have adequately addressed my comments; I support its publication.

Response to all reviewers: We are grateful to the reviewers for their insightful and supportive comments, which helped us to reach our goals in the demonstration and universal understanding of ultralow-voltage operation of LEDs.

Notes on non-scientific updates: Supplementary Figure 11 and Supplementary Table 2 from the previous revision have been moved into the main article as Figure 4 and Table 1.